# HIV-1 Gag Binds the Multi-Aminoacyl-tRNA Synthetase Complex via the EPRS Subunit

**DOI:** 10.3390/v15020474

**Published:** 2023-02-08

**Authors:** Danni Jin, Yiping Zhu, Heidi L. Schubert, Stephen P. Goff, Karin Musier-Forsyth

**Affiliations:** 1Department of Chemistry and Biochemistry, Center for Retrovirus Research, Center for RNA Biology, Ohio State University, Columbus, OH 43210, USA; 2Departments of Biochemistry and Molecular Biophysics, and Microbiology and Immunology, Columbia University Medical Center, New York, NY 10032, USA; 3Department of Microbiology and Immunology, University of Rochester Medical Center, Rochester, NY 14642, USA; 4Department of Biochemistry, University of Utah, Salt Lake City, UT 841122, USA

**Keywords:** human immunodeficiency virus type 1, multi-aminoacyl-tRNA synthetase complex, glutamyl-prolyl tRNA synthetase, HIV-1 MA–tRNA interactions, HIV-dependency factors

## Abstract

Host factor tRNAs facilitate the replication of retroviruses such as human immunodeficiency virus type 1 (HIV-1). HIV-1 uses human tRNA^Lys3^ as the primer for reverse transcription, and the assembly of HIV-1 structural protein Gag at the plasma membrane (PM) is regulated by matrix (MA) domain–tRNA interactions. A large, dynamic multi-aminoacyl-tRNA synthetase complex (MSC) exists in the cytosol and consists of eight aminoacyl-tRNA synthetases (ARSs) and three other cellular proteins. Proteomic studies to identify HIV–host interactions have identified the MSC as part of the HIV-1 Gag and MA interactomes. Here, we confirmed that the MA domain of HIV-1 Gag forms a stable complex with the MSC, mapped the primary interaction site to the linker domain of bi-functional human glutamyl-prolyl-tRNA synthetase (EPRS), and showed that the MA–EPRS interaction was RNA dependent. MA mutations that significantly reduced the EPRS interaction reduced viral infectivity and mapped to MA residues that also interact with phosphatidylinositol-(4,5)-bisphosphate. Overexpression of EPRS or EPRS fragments did not affect susceptibility to HIV-1 infection, and knockdown of EPRS reduced both a control reporter gene and HIV-1 protein translation. EPRS knockdown resulted in decreased progeny virion production, but the decrease could not be attributed to selective effects on virus gene expression, and the specific infectivity of the virions remained unchanged. While the precise function of the Gag–EPRS interaction remains uncertain, we discuss possible effects of the interaction on either virus or host activities.

## 1. Introduction

The majority of anti-retroviral drugs target viral factors and are at risk of becoming ineffective due to the high mutation rate of virus replication and the emergence of drug-resistant viruses. Human immunodeficiency virus type 1 (HIV-1) (for a list of abbreviations) exploits host cell factors to complete nearly every step of its replication cycle [1,2,3,4,5,6,7,8,9]. Targeting these non-viral HIV-dependency factors (HDFs) is a promising therapeutic strategy, as such drugs offer a higher genetic barrier to the selection of drug-resistant viral variants than conventional anti-retrovirals [10].

Among the numerous host cell HDFs are tRNAs and aminoacyl-tRNA synthetases (ARSs) (for a recent review, see ref. [11] and figures therein). HIV-1 uses cellular tRNA as the primer to initiate reverse transcription of the genomic RNA (gRNA) [12]. The highly structured 5′ untranslated region (UTR) of HIV-1 gRNA contains an 18-nucleotide (nt) primer binding site (PBS) complementary to the 3′ end of human tRNA^Lys3^ [13]. The primer tRNA^Lys3^ is selectively packaged into the virion during HIV-1 assembly along with the other major tRNA^Lys^ isoacceptor, tRNA^Lys1,2^ [14,15]. Human lysyl-tRNA synthetase (LysRS) is the only cellular factor known to specifically bind to all tRNA^Lys^ isoacceptors and plays a non-canonical role in primer recruitment into HIV-1 particles [16]. LysRS is normally present in a multi-aminoacyl-tRNA synthetase complex (MSC) in the cytoplasm. Upon HIV-1 infection, LysRS is phosphorylated and undergoes a conformational change, dislodging it from the MSC. Free LysRS interacts with Gag and/or GagPol, and is packaged into the immature viral particle together with all tRNA^Lys^ isoacceptors [14,17,18,19]. The process of annealing the 3′ ends of a subset of these virion-associated tRNA^Lys^ molecules to the PBS is complex and facilitated by the nucleic acid chaperone function of the nucleocapsid domain of Gag [20].

In addition to functions in protein translation and priming of retroviral reverse transcription, cellular tRNAs are also involved in regulating HIV-1 Gag assembly at the plasma membrane (PM) [21,22]. The Gag polyprotein consists of the myristoylated membrane-binding matrix (MA) domain, the capsid (CA) domain, which mediates Gag oligomerization and assembly, the nucleocapsid (NC), which facilitates gRNA packaging, tRNA primer annealing and reverse transcription, and the C-terminal p6 domain, which plays a role in viral budding from the host cell [22]. The MA domain of Gag encodes a lysine/arginine-rich highly basic region (HBR), which interacts with negatively charged lipid membranes [23,24,25]. The MA HBR region binds specifically to phosphatidylinositol-(4,5)-bisphosphate [PI(4,5)P_2_], an anionic lipid enriched at the inner leaflet of the PM; this lipid targets Gag assembly to the PM and prevents promiscuous Gag binding to intracellular membranes [26,27]. MA has also been shown to bind RNAs, and this interaction is also mediated by the HBR [28,29]; its preferred cellular RNA target is tRNAs [30,31]. Gag binding to non-PI(4,5)P_2_ membrane lipids in vitro is significantly reduced in the presence of tRNAs [29,32,33]. Initial studies testing a small subset of in vitro transcribed tRNAs showed that tRNA^Pro^ bound Gag more specifically and inhibited Gag membrane binding in vitro more efficiently than tRNA^Lys3^ [32]. In contrast, a more recent study showed that tRNA^Lys3^ was more effective in inhibiting MA–membrane interaction than tRNA^Pro^ and that this capability depended on the tRNA^Lys3^ D-arm [33]. A co-crystal structure between HIV-1 MA and tRNA^Lys3^ revealed the primary interaction site is at the tRNA elbow region that includes the D-loop [34]. Whether specific tRNAs or all tRNAs are capable of regulating Gag-membrane association in cells is unknown, and it remains unclear how Gag recruits tRNAs in the cytosol.

In addition to LysRS (also known as KARS), the human MSC consists of seven other ARSs: glutamyl-prolyl-tRNA synthetase (EPRS), methionyl-tRNA synthetase (MARS), leucyl-tRNA synthetase (LARS), isoleucyl-tRNA synthetase (IARS), aspartyl-tRNA synthetase (DARS), glutaminyl-tRNA synthetase (QARS), and arginyl-tRNA synthetase (RARS); and three scaffold proteins: ARS-interacting multifunctional protein (AIMP) 1/p43, AIMP2/p38, and AIMP3/p18 [35]. The MSC is proposed to serve as a depot for ARSs with novel functions unrelated to tRNA charging [36,37]. Indeed, components of the MSC such as LysRS are exploited by HIV-1 during infection [18]. Moreover, host factor screens revealed that the majority of proteins within the MSC are part of the HIV-1 Gag and MA interactomes [2,8]. Validation of these proposed interactions has not yet been carried out and their physiological importance is unclear.

The bifunctional EPRS is a component of the MSC with well-characterized noncanonical anti-inflammation and anti-viral functions [38,39]. The two ARS catalytic domains of EPRS are connected by a linker region that contains three helix-turn-helix WHEP domains [named after ARSs initially found to bear this domain, tryptophanyl-tRNA synthetase (WARS), histidyl-tRNA synthetase (HARS) and EPRS]. In response to interferon-γ (IFN-γ) stimulation, EPRS is doubly phosphorylated and released from the MSC. Phosphorylated EPRS is directed to the IFN-γ-activated inhibition of translation (GAIT) complex, which binds to a structural element within the 3′-UTR of mRNAs encoding proteins involved in the inflammatory response, inhibiting their translation [38,40,41,42,43]. RNA virus infection triggers a different EPRS phosphorylation event that redirects it to perform an immunomodulatory function. In this role, EPRS promotes the innate immune response mediated by the RIG-I-like receptor pathway, which induces the production of antiviral cytokines and inhibits the replication of RNA viruses [39].

In this study, we confirmed the interaction between the MA domain of HIV-1 Gag with the MSC and identified the EPRS WHEP domains as a primary site of interaction. We characterized the interaction between EPRS and HIV-1 MA in vitro and in cells and explored the potential role of EPRS in HIV-1 replication. Our data suggest that Gag interaction with the MSC may not be limited to the EPRS linker region. We discuss possible mechanisms by which HIV-1 may utilize EPRS for efficient replication, and by which EPRS may mediate antiviral responses to HIV-1 infection. Drugs designed to prevent these HDF interactions may offer promising new anti-viral strategies.

## 2. Materials and Methods

### 2.1. Plasmids, Protein Expression and Purification

Plasmids pcDNA3.EPRS-FLAG and pET30 vectors encoding EPRS linker constructs (whole linker, R1R2R3, R1R2, and R2R3) were generously provided by Dr. Paul Fox (Cleveland Clinic). Site-directed ligase-independent mutagenesis [44,45] was used to generate pcDNA3.EPRS-FLAG with WHEP domain deletions (ΔR1, ΔR3, ΔR1R2, ΔR2R3 and ΔR1R2R3). Viral vector pNL4-3 E-R+ (with a luciferase gene inserted in the place of the viral *Nef* gene and the viral *Env* gene deleted) was obtained from the NIH AIDS reagent program. The glycoprotein of vesicular stomatitis virus (VSV-G) was expressed using pMD2.G, and packaging vector psPAX2 was used for virus production. The empty retroviral vector (pQCXIP-FH) and retroviral vector expressing HIV-1 MA (pQCXIP-MA-FH) have been described previously [46]. For alanine scanning mutagenesis, residues of MA were mutated to alanines in blocks of three. All the mutations were introduced to pQCXIP-MA-FH by overlap PCR. The DNA sequences encoding domains of EPRS (EARS, linker, and PARS) were cloned to pCMV-myc (Clontech) for the expression of myc-tagged ERS, linker, and PRS.

His-tagged HIV-1 MA [32] and EPRS linker constructs [40] were purified using nickel-affinity purification methods adapted from previously described protocols. Briefly, proteins were expressed in *E. coli* BL21(DE3) with 0.1 mM isopropyl β-D-1-thiogalactopyranoside (IPTG) induction at 16 °C overnight. Cells were lysed by sonicating in lysis buffer [20 mM Tris-HCl pH 8, 500 mM NaCl, 10 mM imidazole, 10% glycerol, and 1 mM dithiothreitol (DTT)] supplemented with protease inhibitor cocktail (Roche). To remove nucleic acids, lysates were incubated with 0.5% *v/v* polyethyleneimine (PEI); proteins were precipitated with ammonium sulfate (60% saturation) and resuspended in lysis buffer. Proteins were purified with His-Select Nickel affinity resin (Millipore, Burlington, MA, USA), washed, and eluted with lysis buffer containing a step gradient of imidazole. Elution fractions were analyzed by SDS-PAGE, and fractions containing the protein of interest were combined and dialyzed in dialysis buffer (40 mM HEPES pH 7.5, 150 mM NaCl, and 2 mM DTT). Proteins were concentrated and stored in dialysis buffer at −80 °C, at approximately 100–200 μM. Protein concentrations were measured using the Pierce BCA protein assay kit (Thermo Fisher, Waltham, MA, USA).

### 2.2. RNA In Vitro Transcription and Labeling

Plasmids encoding human tRNA^Pro^, tRNA^Lys3^, tRNA^Glu^, or bovine tRNA^Trp^ under a T7 promotor with *Fok*I (tRNA^Lys3^ and tRNA^Glu^) or *Bst*NI (tRNA^Pro^ and tRNA^Trp^) digestion sites at the 3′ end were linearized and used as templates for in vitro transcription with T7 RNA polymerase as previously described [47]. tRNAs were purified on a 12% denaturing polyacrylamide gel, eluted with buffer containing 0.5 mM NH_4_OAc and 1 mM EDTA, and concentrated using butanol extraction and ethanol precipitation. The tRNAs were labeled with fluorescein-5-thiosemicarbazide (FTSC) at the 3′ end, as previously described [48].

### 2.3. Fluorescence-Quenching and Fluorescence Anisotropy (FA) Assays

The FTSC-labeled tRNAs were folded in 50 mM HEPES pH 7.5 by incubating at 80 °C for 2 min, 60 °C for 2 min, adding 1 mM MgCl_2_, and incubating on ice for a minimum of 30 min. Folded RNAs (5 nM) were incubated with serially diluted EPRS linker proteins (0–18 μM) for 30 min in 20 mM Tris-HCl pH 8, 50 mM NaCl, and 1 mM MgCl_2_, prior to measuring fluorescence intensity. Dissociation constants, *K*_d_, were obtained by plotting fluorescence intensity vs. protein concentration and fitting data to the equation $I\left(P\right)=I_{max}-\Delta I\times \frac{P/K_{d}}{1+P/K_{d}}$, where *I_max_* is the maximum fluorescence intensity, ΔI is the difference between the maximum and minimum fluorescence intensity, and P is protein concentration [48]. For fluorescence anisotropy assays, serially diluted EPRS linker proteins were incubated with 5 nM FTSC-labeled RNAs or 10 nM Alexa Fluor 488-labeled HIV-1 MA protein for 30 min in the same Tris buffer as above. Data were analyzed as previously described [48], and *K*_d_ values were derived from three independent experiments.

### 2.4. Cell Culture and Stable Cell Line Generation

HEK293, HEK293T, and HeLa cells were cultured in Dulbecco’s Modified Eagle’s Medium (DMEM) supplemented with 10% (*v*/*v*) fetal bovine serum (FBS), 100 IU/mL penicillin, and 100 μg/mL streptomycin (complete DMEM). Transformed human T cells expressing C-C chemokine receptor 5 (HuT/CCR5) were cultured in complete Roswell Park Memorial Institute (RPMI) medium containing 1 μg/mL puromycin and 500 μg/mL geneticin. GHOST X4/R5 reporter cells were cultured in complete DMEM with 1 μg/mL puromycin, 500 μg/mL geneticin, and 100 μg/mL hygromycin B. For the generation of inducible shRNA-expressing cell lines, pTRIPZ plasmids encoding a doxycycline-inducible EPRS-specific shRNA (clone ID V3THS_396942, sequence 5′-AGTTGTATAGTCTCCTCCT-3′) or a proprietary non-silencing shRNA were purchased from Dharmacon and used to generate stable cell lines by lentivirus transduction following the manufacturer’s protocol. To generate a stable cell line that inducibly expresses FLAG-tagged EPRS linker, the FLAG-linker coding sequence was cloned into pTRIPZ vectors between *Age*I and *Mlu*I restriction sites, and the resulting plasmid was used for lentivirus transduction as described above. The stable cell lines were selected and maintained in complete DMEM with 1 μg/mL puromycin. For inducing shRNA or FLAG-linker expression, cells were kept in media containing 1.5 μg/mL doxycycline throughout the duration of the experiment. HEK293 cells stably transduced with empty retroviral vector (MA-FLAG-) or retroviral vector expressing flag-tagged MA (MA-FLAG +) have been described previously [46].

### 2.5. Immunoprecipitation (IP) and Western Blotting

For FLAG-MA IPs, stably transduced HEK293 cells or cells transfected with WT or mutant pQCXIP-MA-FH were lysed in CelLytic M Cell Lysis Reagent (Sigma, Tokyo, Japan, C2978) for 10 min. The lysate was clarified by centrifugation at 4 °C for 15 min at 12,000 rpm. The supernatant was mixed with ANTI-FLAG M2 Affinity Gel (Sigma, A2220), and the mixture was incubated at 4 °C for 4 h. The resin was washed with TBST (TBST) four times, and the proteins bound to the resin were recovered and resolved by SDS-PAGE electrophoresis, transferred to a PVDF membrane, and probed by Western blotting. To identify MA-interacting proteins, immunoprecipitated samples were analyzed by SDS-PAGE, and individual bands were submitted for mass spectrometry analysis at the Columbia University Medical Center protein core facility.

To test the interaction between HIV-1 Gag and full-length or truncated EPRS in cells, 1.5 × 10^6^ HEK293T shEPRS-inducible cells were seeded on 10 cm dishes, and 1.5 μg/mL doxycycline was added to the media the following day to induce shRNA expression. Cells were co-transfected with 10 μg full-length or truncated pcDNA3.EPRS-FLAG and 10 μg pGag-GFP 24 h after doxycycline treatment using the PEI method [49]. Cells were lysed 48 h post-transfection in cell-lysis buffer (Cell Signaling Technology, Danvers, MA, USA) containing a protease inhibitor cocktail (Sigma-Aldrich). For IP experiments, 5 μg FLAG M2 mouse monoclonal antibody (Sigma-Aldrich F1804) or mouse IgG control (Invitrogen 10400C, Waltham, MA, USA) was conjugated to 25 μL Dynabeads protein G (Invitrogen). Cell lysate (300 μL) containing approximately 1200 μg total protein was applied to antibody-conjugated beads and incubated at 4 °C overnight. Beads were washed 3 times with 0.1% Tween 20 in phosphate-buffered saline (PBS) and eluted with SDS-PAGE loading buffer by heating at 95 °C for 5 min, followed by Western blotting.

Antibodies used in Western blot (Figures 2, 3A, and 5) were specific for: FLAG (Sigma, F1804); Myc (Santa Cruz, Santa Cruz, CA, USA, sc-40); RARS (Abcam, Cambridge, UK, ab31537); QARS (Abcam, ab72957); EPRS (Abcam, ab31531); LARS (Abcam, ab31534); MARS (Abcam, ab 31541); LARS (Abcam, ab 31533); IARS (Abcam, ab31533); p18 (Abcam, ab31543); KARS (Abcam, ab129080); DARS (Abcam, ab182157); p38 (Abcam, ab228004); p43 (Abcam, ab188320). Antibodies used in Western blot (Figures 3C, 6–8) were: FLAG (Sigma, F1804), EPRS (Novus Biologicals, Englewood, CO, USA, NBP1-84929), HIV-1 p24 (Invitrogen, MA1-7040), and GAPDH (Bio-Rad, Hercules, CA, USA).

### 2.6. Streptavidin Pull-Down Assay

Plasmid pET151-His10-PP-EPRS (CPH2944) encoding residues 683–1023 of the EPRS linker domain was expressed in 2 L autoinduction media [50] at 37 °C to an OD600 of 0.6 and then overnight at 19 °C. The pellet was lysed by sonication in 20 mM Tris pH 7.5, 300 mM NaCl, 40 mM imidazole, 5% glycerol with added protease inhibitors (pepstain, leupeptin, and aprotinin) and clarified. The supernatant was incubated with 10 mL pre-equilibrated Ni2+ resin (Qiagen, Hilden, Germany) for 30 min prior to washing with 100 mL lysis buffer and 50 mL wash buffer (same as lysis with 100 mM NaCl and no protease inhibitors). The protein was eluted with 6 × 8 mL fractions of elution buffer (20 mM Tris pH 7.5, 100 mM NaCl, 250 mM Imidazole, 5% glycerol) and dialyzed overnight into 20 mM Tris pH 7.5, 100 mM NaCl, 1 mM DTT, 5% glycerol). Anion exchange chromatography was performed, and the protein was eluted using a gradient of 100 mM–1 M NaCl. Positive fractions were pooled and concentrated prior to size-exclusion chromatography over a 26/60 Superdex 200 column.

Plasmid pET11a-MA-PP-Strep (CPH2952) encoding Strep-tagged HIV-1 MA was expressed in 2 L autoinduction media as described above. The pellet was lysed by sonication in 20 mM Tris pH 7.5, 300 mM NaCl, 10% glycerol, and 1 mM DTT. The clarified lysate was incubated with equilibrated Strep resin and washed with wash buffer (20 mM Tris pH 7.5, 150 mM NaCl, 10% glycerol, and 1 mM DTT) prior to separation into experimental fractions and incubation with no protein or EPRS linker with or without 2 µL of 25 mg/mL RNase1 (Qiagen) for 2 h at 4 °C with rocking. Additional resin was incubated only with EPRS linker (683–1023) as a control. After incubation, the resin was washed with 5 column volumes of buffer (20 mM Tris pH 7.5, 150 mM NaCl, 5% glycerol, and 1 mM DTT), and samples of resin were removed for analysis by SDS-PAGE and visualized by Coomassie staining.

### 2.7. Immunofluorescence Microscopy

For immunofluorescence microscopy analyses of HIV-1-infected HeLa cells, 5 × 10^4^ cells were seeded on poly-L-lysine treated coverslips a day prior to transfection. Cells were infected with HIV-Luc/VSV-G at an MOI of 1 for 2 h. Cells were fixed 24 h post-infection with 4% paraformaldehyde at room temperature for 20 min and permeabilized using 0.1% Triton X-100. Cells were blocked with 5% bovine serum albumin dissolved in PBS before treatment with each of the following antibodies at room temperature for 1 h: rabbit anti-EPRS (Bethyl A103-957A, 1:250 dilution in 2% FBS/PBS), Dylight 550 goat anti-rabbit (ThermoFisher, 1:500 dilution), mouse anti-HIV-1 p24 (Invitrogen MA1-7040, 1:100 dilution), and Dylight 488 goat anti-mouse (ThermoFisher, 1:500 dilution). Nuclei were stained with DAPI (Molecular Probes, Eugene, OR, USA, 1:40,000 dilution in PBS). Cells were washed with 2% FBS in PBS in between and after antibody and DAPI treatment. Deconvolution images were obtained using a DeltaVision microscope (GE, Boston, MA, USA) with an oil immersion (60×/NA 1.4) objective lens.

### 2.8. Virus Production and Infectivity Assays

To produce HIV-Luc/VSV-G pseudotyped virus, 3.5 × 10^6^ HEK293T producer cells were transfected with 10 μg pNL4-3Luc E-R+ viral plasmid and 2 μg pMD2.G (encoding VSV-G). Virus-containing supernatant was collected 48 h post-transfection and filtered through a 0.45 μm syringe filter. The titer of the virus was measured by limiting dilution on GHOST X4/R5 reporter cells as previously described [51]. M7, M8, M9, M11, and M26 mutations were introduced into pNL4-3Luc E-R- viral plasmids, and mutant viruses were generated in a similar way. To determine the relative infectivity of progeny virions produced from EPRS knockdown cells, stable cell lines expressing EPRS-specific or control shRNAs were treated with doxycycline a day prior to transfection, and viruses were produced and titered as described above. The concentration of viral CA/p24 was measured by enzyme-linked immunosorbent assay (ELISA) using an HIV-1 p24 antigen ELISA kit (ZeptoMatrix) according to the manufacturer’s protocol. Relative infectivity was calculated as infectious unit per pg of p24.

The supernatant medium from cells (3 mL) was layered above 1 mL of 25% sucrose in TEN buffer [10 mM Tris-Cl (pH 8.0), 0.1 M NaCl, and 1 mM EDTA (pH 8.0)]. Samples were centrifuged at 100,000× *g* (~28,000 rpm) for 2 h at 4 °C (SW55 rotor, Beckman, Brea, CA, USA). The virus-like particle pellets were resuspended in 100 μL of 1 × SDS loading buffer, resolved by SDS-PAGE, and analyzed by Western blot.

### 2.9. Quantitative Real-Time (qRT)-PCR and Luciferase Assays

To characterize the effect of EPRS overexpression on HIV-1 gene expression, 5 × 10^5^ HEK293T cells transfected with 2 μg pcDNA3.EPRS-FLAG or empty vector were infected with HIV-Luc/VSV-G at an MOI of 1. To test the effect of EPRS knockdown, shRNA-expressing stable cells were treated with 1.5 μg/mL doxycycline 24 h prior to HIV-Luc/VSV-G infection at an MOI of 2.5. As an internal control, cells were transfected with the Renilla luciferase-expressing plasmid pRL-TK immediately after infection. Cells were lysed 24 h post-infection for luciferase assays or qPCR. For luciferase assays, cells were lysed with reporter lysis buffer (Promega, Madison, WI, USA) and luciferase activity was measured using the Dual-Luciferase^®^ Reporter Assay System (Promega) following the manufacturer’s protocol. For qRT-PCR, total RNA was extracted from cell pellets using Aurum™ Total RNA Mini Kit (Bio-Rad); 50 ng was analyzed using the iTaq™ Universal SYBR Green One-Step Kit (Bio-Rad) and the following primers: Firefly luciferase (forward: 5′-GGTTGGCAGAAGCTATGAAACG-3′, reverse: 5′-CATTATAAATGTCGTTCGCGGG-3′); spliced GAPDH: (forward: 5′-GGAAGGTGAAGGTCGGAGTCAACGG-3′, reverse: 5′-CTGTTGTCATACTTCTCATGGTTCAC-3′).

## 3. Results

### 3.1. HIV-1 MA Interacts with the MSC Primarily through EPRS

Published affinity purification followed by mass spectrometry studies revealed that the interactome of HIV-1 MA protein in HEK293 and Jurkat cells includes all components of the MSC [2]. To validate the interaction of HIV-1 MA with the MSC, we generated stable HEK293 cell lines expressing FLAG-tagged HIV-1 MA or the parental empty vector and performed IP assays with FLAG antibody. Proteins co-immunoprecipitated with MA-FLAG were analyzed by SDS-PAGE, visualized by Coomassie blue staining (Figure 1), and identified by mass spectrometry. The highest molecular weight bands (1–6) corresponded to six ARSs, which are all known components of the MSC (EPRS, IARS, LARS, MARS, QARS, and KARS). Band 7 had five components, with the majority of peptides corresponding to elongation factor-1α and Obg-like ATPase 1 (OLA1) [2]. Band 8 contained two known MSC scaffold proteins (AIMP1/2) as the major components. In a follow-up experiment, the presence of these interacting MSC components was confirmed by immunoblotting, where all MSC components tested were shown to coimmunoprecipitate with MA-FLAG (Figure 2, lanes 1 and 2). These observations suggest that the MSC constitutes the most abundant MA-interacting proteins in the host that are recovered by coimmunoprecipitation.

To identify the direct interacting partners of MA within the MSC, cells expressing MA-FLAG were transfected with siRNAs targeting individual MSC proteins or with a non-targeting siRNA. Knockdown of MSC components was confirmed by immunoblotting (Figure 2, top panel). FLAG co-IP experiments performed in the presence of specific siRNAs showed that knockdown of MSC scaffold proteins p43, p38, and p18 disrupted the interaction between MA and some MSC proteins, whereas knocking down EPRS reduced the MA interaction with almost all MSC components (Figure 2, bottom panel). These data support a key role for EPRS in the MA–MSC interaction.

**Figure 1 viruses-15-00474-f001:**
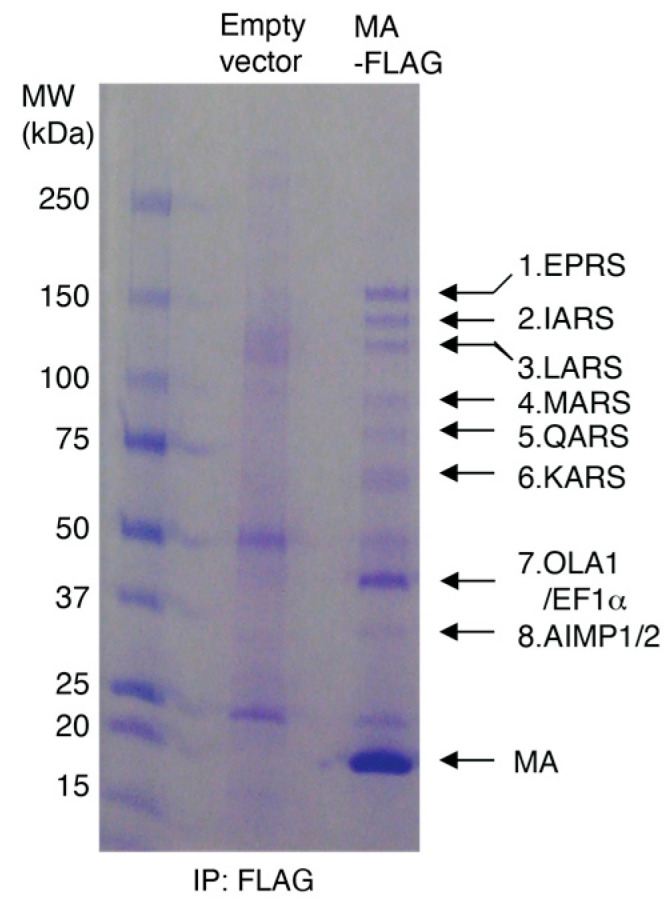
Co-purification of HIV-1 MA with MSC components. Lysates of HEK293 cells expressing MA-FLAG or empty vector control were subjected to FLAG immunoprecipitation. The precipitated protein was analyzed by SDS-PAGE and visualized by Coomassie blue staining. MSC components that co-immunoprecipitated with MA are indicated.

**Figure 2 viruses-15-00474-f002:**
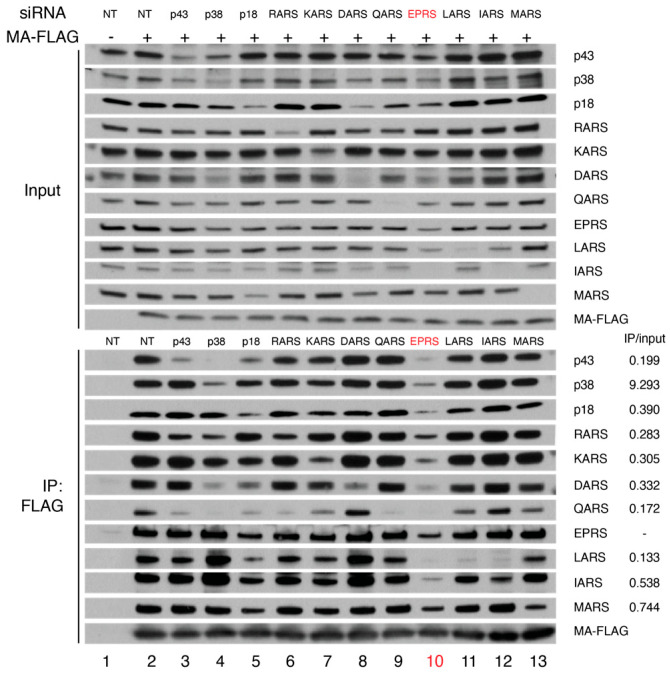
FLAG-IP in the absence or presence of siRNA knockdown of MSC proteins. Immunoprecipitation was performed using FLAG antibody in HEK293 cells without (lane 1) or with (lane 2) expression of MA-FLAG in the presence of a non-targeting (NT) siRNA. Lanes 3–13: siRNAs targeting each MSC component were expressed, and FLAG-IP was carried out for each knockdown. Top shows input signals indicating successful knockdowns, and bottom shows the results of the FLAG-IP. IP versus input ratios (siEPRS relative to NT) are labeled on the bottom right.

### 3.2. HIV-1 MA Interacts with the Linker Region of EPRS in an RNA-Dependent Manner

To investigate the domains of EPRS required for MA interaction, we expressed myc-tagged EPRS fragments (EARS, linker, or PARS) in MA-FLAG-expressing or control HEK293 cells. Only the linker domain fragment was observed to co-IP with MA, suggesting that the linker region but not either of the ARS catalytic domains interacts with MA (Figure 3A). To establish if the interaction was RNA dependent, bacterially expressed EPRS linker and strep-tagged MA protein were incubated with streptavidin beads in the absence or presence of RNaseA. RNaseA treatment abrogated the interaction between MA and EPRS linker, confirming that the interaction was RNA dependent (Figure 3B). The fluorescence anisotropy binding assay performed with purified fluorescently labeled MA and EPRS linker also did not support a direct binding interaction between the recombinant proteins (Appendix A). Taken together, the data suggest that HIV-1 MA interacts with the linker region of EPRS in an RNA-dependent manner.

Based on a proteomics study, similar to MA, the full-length Gag polyprotein also interacted with ARSs within the MSC [8]. A size-exclusion chromatography analysis of HIV-1-infected cell lysate also showed that Gag co-eluted with MSC components EPRS, LysRS, LeuRS, and AIMP2/p38, supporting a Gag interaction with the MSC [18]. To test the importance of the linker WHEP domains in EPRS interactions with full-length Gag, we expressed EPRS-FLAG and Gag-GFP in HEK293T cells after removing the endogenous EPRS with EPRS-specific shRNA knockdown (Figure 3C). Immunoprecipitation with FLAG antibody showed that Gag co-precipitated with EPRS-FLAG (Figure 3C, bottom, lane 2). EPRS-FLAG constructs with a variety of WHEP domain deletions were generated and tested for their ability to co-IP with Gag (Figure 3C, top). EPRS constructs with one (ΔR1 and ΔR3), two (ΔR1R2 and ΔR2R3), or all three WHEP domains deleted (ΔR1R2R3) maintained their ability to interact with Gag (Figure 3C, lanes 3–7). One possible explanation is that the interaction between EPRS and Gag does not require the helical WHEP domains but rather the random coiled portion of the linker domain, which was present in all constructs (Figure 3C, top gray).

**Figure 3 viruses-15-00474-f003:**
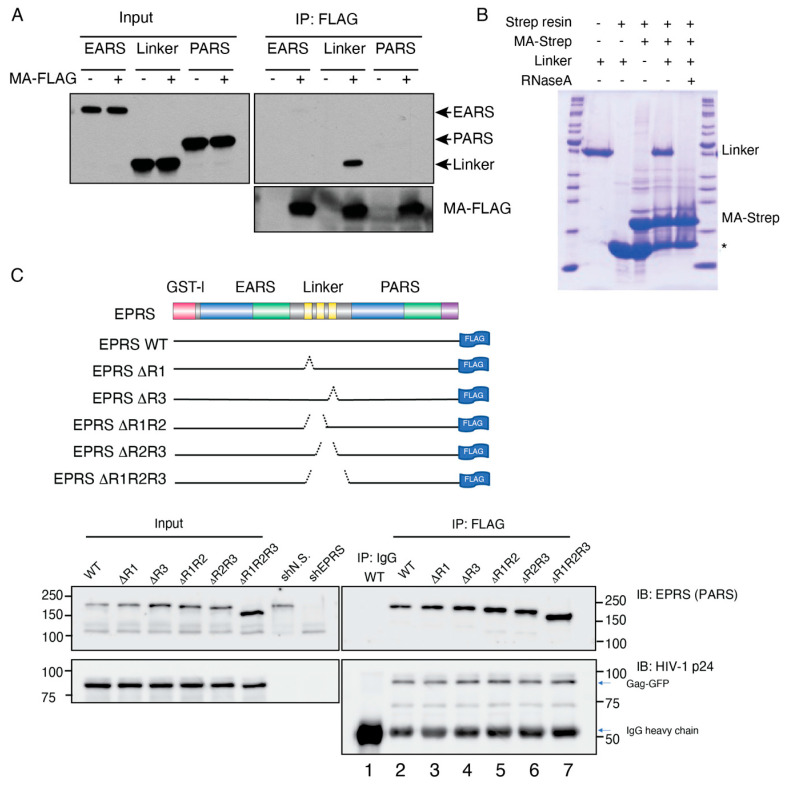
Domain mapping of MA–EPRS interaction. (**A**) Individual domains of EPRS (EARS, linker, and PARS) were co-expressed with MA-FLAG and subjected to co-immunoprecipitation with FLAG antibody. (**B**) Bacterially expressed EPRS linker and MA-strep protein were incubated with strep beads with or without RNaseA treatment. The proteins bound to the beads were resolved on an SDS-PAGE gel and visualized by Coomassie staining. Band with asterisk (*) indicates streptavidin monomer eluted from the strep affinity resin. (**C**) Top: schematic of WT EPRS and truncated Δlinker constructs. One, two, or all three WHEP domains (yellow bars in the linker region) were truncated. Bottom: co-IP of HIV-1 Gag-GFP with EPRS-FLAG (full-length WT or with WHEP domain truncations, as indicated). EPRS-FLAG constructs were co-expressed with Gag-GFP in an EPRS knockdown background.

### 3.3. The Linker Domain of EPRS Interacts Promiscuously with tRNAs

While RNA is required for HIV-1 MA interaction with EPRS (Figure 3B), whether a specific RNA species mediates the interaction is unclear. A crosslinking-IP sequencing study showed that tRNAs are the most frequent Gag-bound RNAs in cells and that the MA is domain bound almost exclusively to tRNAs [30]. Given that tRNAs are the native substrates of ARSs and are presumably enriched near the MSC, we hypothesized that the MA–EPRS interaction is mediated by tRNAs. To test this hypothesis, we first investigated the tRNA binding properties of the EPRS linker using purified recombinant linker-derived proteins and four in vitro transcribed, 3′ fluorescently labeled tRNAs. We tested binding to human tRNA^Lys3^, the primer for HIV-1 reverse transcription, human tRNA^Glu^, and tRNA^Pro^, the substrates of EPRS, and bovine tRNA^Trp^ as an unrelated control. EPRS linker proteins were titrated into 5 nM of each tRNA, and the dissociation constant, *K*_d_, was derived from fluorescence quenching curves (Figure 4). The full-length EPRS linker displayed similar sub-micromolar affinities for all tRNAs tested (Table 1). Thus, the linker region of EPRS does not distinguish between tRNA species in vitro.

In previous studies, the three individual WHEP domains displayed different RNA binding properties even though they have similar helix-turn-helix structures [40]. To characterize the contribution of individual WHEP domains to tRNA binding, we purified EPRS linker fragments containing two or three WHEP domains (R1R2, R2R3, and R1R2R3, see Figure 4B) and tested the tRNA^Lys3^ binding affinity of each protein via FA assays (Figure 4C). R1R2R3 and R1R2 displayed sub-micromolar affinity for tRNA^Lys3^, whereas R2R3 failed to result in significant binding (Figure 4C and Table 1). We conclude that the upstream WHEP domains (R1R2) are the major contributor to the affinity of the linker domain for tRNA. This result is in good agreement with a previous report showing that the hairpin structure within the GAIT mRNA binds to R1R2 with a higher affinity compared to R2R3 [40].

### 3.4. Identification of Amino Acid Residues in MA Critical for EPRS Interaction

To identify the region of MA involved in the interaction with EPRS, we performed alanine-scanning mutagenesis. Forty-three MA mutants were generated (M1-M43), each having three adjacent amino acids mutated to alanine (Figure 5A, top). FLAG-tagged MA mutants were expressed in HEK293T cells and tested for EPRS interaction via co-IP. Five of the mutant constructs (M7, M8, M9, M11, and M26) displayed significant defects in EPRS interaction despite robust MA expression (Figure 5A, bottom). The residues critical for EPRS interaction were mapped on a previously reported structure of HIV-1 MA bound to a PI(4,5)P_2_ analog [52]. Interestingly, all the critical residues are located proximal to the PI(4,5)P_2_ binding site on the same face of MA (Figure 5D). We conclude that MA interacts with EPRS via the same surface as it interacts with PI(4,5)P_2_ at the PM.

**Figure 5 viruses-15-00474-f005:**
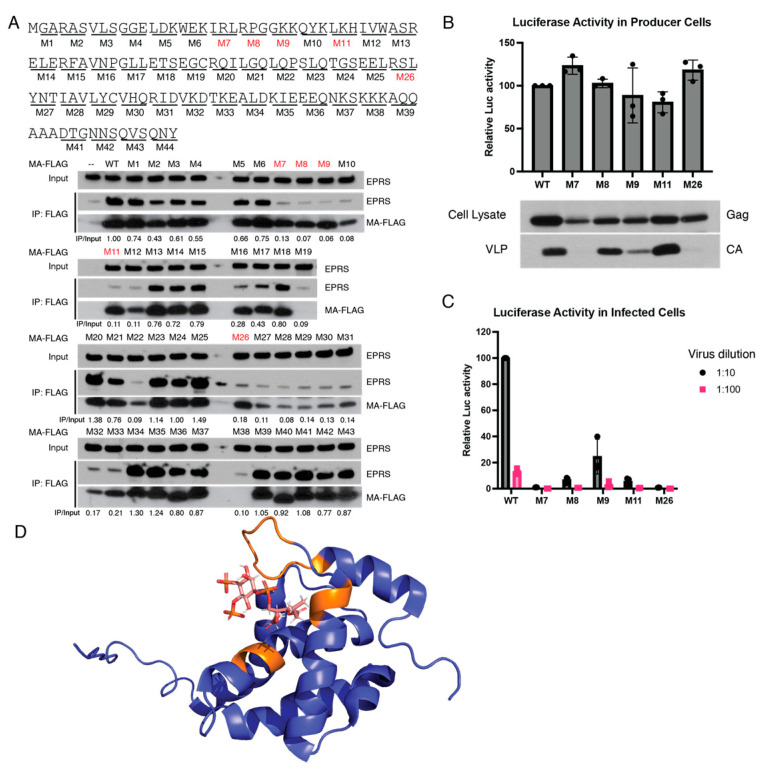
Identification of MA residues critical for interaction with EPRS. (**A**) Alanine-scanning mutagenesis screen to identify interaction-ablative MA mutants. MA residues were mutated to alanine in groups of three. A total of 43 mutants were generated and tested for EPRS interaction via co-IP. Five mutants with robust MA expression but significantly reduced EPRS interaction are indicated with red boxes. EPRS IP/input ratio (normalized to WT) is labeled below each lane. Mutations of interest were selected based on two criteria: (1) low IP/input ratio; (2) MA-FLAG expression is comparable to WT. (**B**) Production of progeny virions by MA mutants as measured via luciferase activity in producer cells (**top**). Gag expression in producer cells and HIV-1 CA in supernatants as assessed by immunoblotting (**bottom**). The top of the bars represents mean value of 3 trials. (**C**) Relative infectivity of virion-containing supernatants measured by luciferase activity in target cells. The top of the bars represents mean value of 3 trials. (**D**) Critical interacting residues (orange) mapped onto a previously reported crystal structure of MA binding to PI(4,5)P_2_ (PDB: 2H3Z) [52].

To examine the effects of these mutations on virus replication, these 5 mutations were introduced into HIV-luciferase viral vectors and tested for progeny virion production and infectivity. The various mutants displayed a complex array of phenotypes. While all mutant Gag proteins were well expressed in producer cells, M7 and M26 mutants were defective for virion production (Figure 5B). The M9 mutant produced reduced levels of virions, while viral vectors encoding M8 and M11 mutants produced a similar amount of progeny virions as WT. These findings suggest that the interaction with EPRS is not strictly essential for virion production. The relative infectivity of the progeny virions was compared by measuring luciferase activity in cells infected with comparable volumes of WT and mutant virion-containing supernatants, not attempting to correct for the virion abundance. M8 and M11 virions exhibited at least a 10-fold reduced infectivity compared to WT. M9 infectivity was also reduced relative to WT, but these virions were significantly more infectious than M8 and M11 given the low concentration of virions in the supernatant of producer cells (Figure 5C). It remains uncertain if these phenotypes can be attributed to the loss of interaction with EPRS or to defects in other MA functions such as the interaction with PI(4,5)P_2_.

### 3.5. Single-Cycle HIV-1 Infection Does Not Alter the Expression of EPRS

To investigate the potential impact of virus on EPRS, we first tested the expression level of EPRS in cells infected with single-cycle HIV-luciferase/vesicular stomatitis virus G (HIV-Luc/VSV-G) pseudotyped virus. Infection of HEK293T and transformed human CD4+ T cells expressing C-C chemokine receptor 5 (HuT/CCR5) was confirmed by immunoblotting for Gag p55 or via luciferase assays with infected cell lysates. EPRS protein levels remained unchanged 24 h post-infection (Figure 6A,B). To observe EPRS expression at the single-cell level, HeLa cells were infected with HIV-Luc/VSV-G, and EPRS and Gag were visualized via immunofluorescence. The EPRS level in Gag-positive infected cells was approximately the same as that in non-infected cells (Figure 6C). This result is consistent with a previous proteo-transcriptome host factor screen in HIV-1-infected SupT1 CD4+ cells, where no changes were observed in EPRS transcript, protein, and phosphoprotein levels relative to uninfected cells [53].

**Figure 6 viruses-15-00474-f006:**
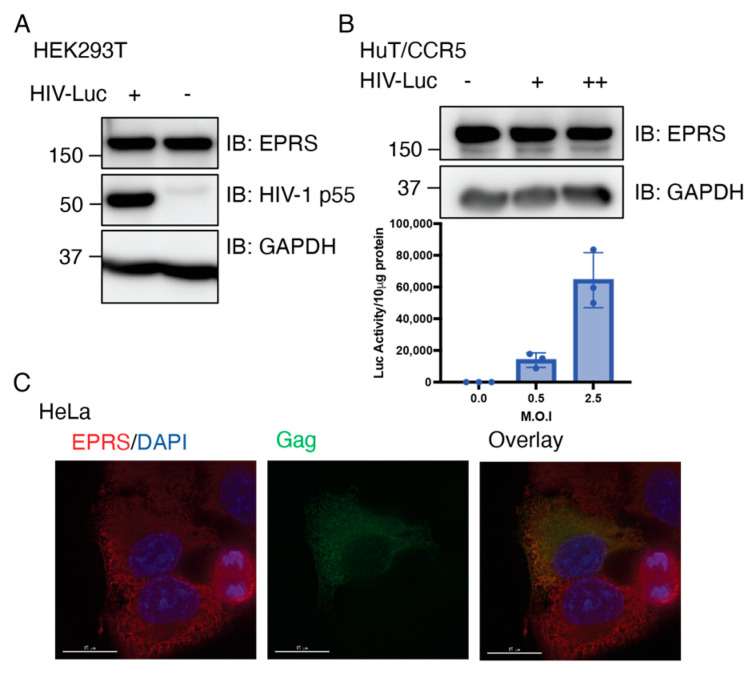
EPRS protein levels are not altered in cells infected with single-cycle HIV-1. (**A**) Immunoblot showing level of EPRS in HEK293T cells uninfected or infected with HIV-Luc/VSV-G with M.O.I. of 5. (**B**) Immunoblot showing level of EPRS in HuT/CCR5 cells uninfected or infected with HIV-Luc/VSV-G with MOI of 0.5 (+) or 2.5 (++). Graph shows level of infectivity as measured by luciferase production. (**C**) Immunofluorescence imaging of Gag-GFP and EPRS expression in HeLa cells infected with HIV-Luc/VSV-G.

### 3.6. EPRS Knockdown Affects HIV-1 Gene Expression Due to Global Translational Defects

To further explore the effect of EPRS on HIV-1 gene expression, HEK293T cells overexpressing full-length EPRS or the linker domain alone were infected with HIV-Luc/VSV-G and assayed for luciferase activity 24 h post-infection. No significant difference was observed in cells overexpressing full-length EPRS or linker domain relative to the vector alone control (Figure 7A). To determine the effect of EPRS knockdown on HIV-1 gene expression, stable HEK293T cell lines that allow doxycycline-inducible expression of EPRS-targeting shRNA (shEPRS) or non-silencing shRNA (shN.S.) were generated (Figure 7B) and subjected to infection by HIV-Luc/VSV-G encoding Firefly (FF) luciferase. As an internal control, cells were simultaneously transfected with a vector encoding Renilla luciferase under the control of an HSV-thymidine kinase (TK) promoter (pRL-TK). A 4-fold decrease in HIV-1 gene expression (indicated by FF luciferase activity) and a similar decrease in non-HIV gene expression (indicated by Renilla luciferase activity) were observed in cells with EPRS knockdown (Figure 7C,D). When FF luciferase activity was normalized to Renilla luciferase activity, similar levels of HIV-1 gene expression were observed between EPRS knockdown and control cells (Figure 7E), suggesting that the negative effect of EPRS knockdown on gene expression was not HIV-specific. The mRNA level of FF luciferase showed no difference between EPRS knockdown and control cells, indicating that EPRS has no specific effect on the transcription of HIV-1 genes (Figure 7F). Instead, EPRS knockdown affects the translation of FF luciferase, likely due to global effects on tRNA aminoacylation.

**Figure 7 viruses-15-00474-f007:**
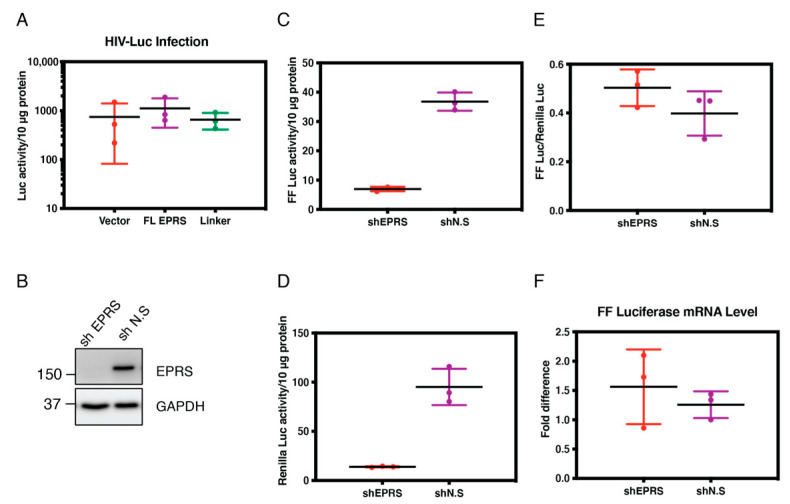
Effect of EPRS on HIV-1 gene expression. (**A**) Relative luciferase activity in HEK293T cells overexpressing full-length (FL)EPRS-flag or linker-flag (as indicated in the Western blot below the graph, probed with anti-flag antibody) and infected by HIV-Luc/VSV-G. (**B**) Stable HEK293T cell lines that allow doxycycline-inducible expression of EPRS-targeting or non-silencing shRNAs (shEPRS and shN.S.) were generated and treated with doxycycline for 72 h. EPRS protein level was measured by immunoblotting. (**C**–**E**) Luciferase activities in cells infected with HIV-Luc/VSV-G and co-transfected with pRL-TK. Firefly (**C**) and Renilla (**D**) luciferase activity was measured by a dual-luciferase assay. (**E**) To assess HIV-specific effects, Firefly luciferase activity was normalized to Renilla luciferase activity. (**F**) Firefly luciferase mRNA levels in cells infected with HIV-Luc/VSV-G, measured by qRT-PCR. Experiments were performed in triplicate, with the mean value indicated by a horizontal line. Data are shown as luciferase mRNA levels relative to one of the shN.S + HIV samples, which was set to 1.0.

### 3.7. EPRS Knockdown Reduces Progeny Virion Production but Not Infectivity

To examine the potential role of EPRS on progeny virion yield and infectivity, shEPRS and shN.S-expressing producer cells were transfected with pNL4-3 E-R+ viral plasmid and VSV-G expressing plasmid pMD2.G. Progeny virion-containing supernatants were collected, measured for p24 levels by ELISA, and titered for virion infectivity in transformed human osteosarcoma (GHOST) indicator cells. Gag production in producer cells was visualized by immunoblotting (Figure 8A). Approximately 2-fold less progeny virions were produced in cells with EPRS knockdown compared to the control cells (Figure 8B). The amount of infectious virions in the supernatant of shEPRS-expressing cells was similarly reduced by 2-fold (Figure 8C). The relative infectivity as indicated by virion titer normalized by p24 concentration exhibited no significant difference regardless of EPRS knockdown in producer cells (Figure 8D). Taken together, EPRS knockdown in producer cells resulted in reduced progeny virion production, whereas virion infectivity was unaffected. To investigate the effect of the linker domain alone on progeny virion production and infectivity, HEK293T cells with doxycycline-inducible expression of the EPRS linker were used as producer cells. Virion titer and p24 release were similar between induced and uninduced cells (Appendix A).

**Figure 8 viruses-15-00474-f008:**
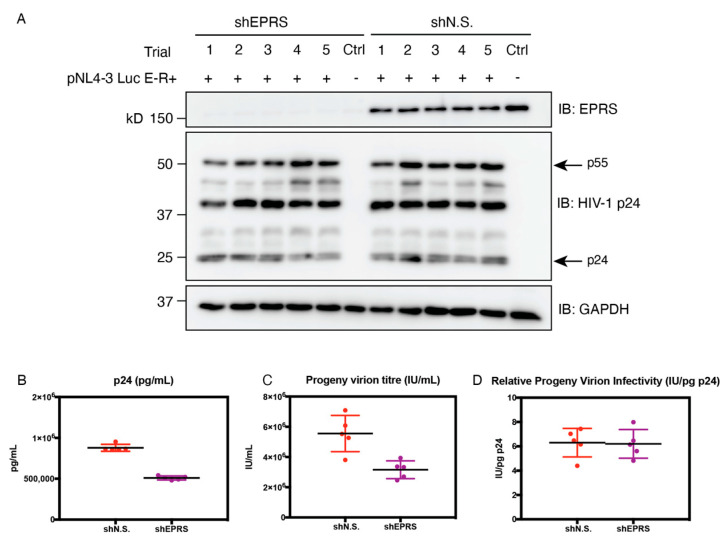
Progeny virion production, titer, and infectivity. HEK293T cells expressing EPRS-targeting or non-silencing shRNA were transfected with pNL4-3 E-R+ and VSV-G-expressing plasmids. (**A**) Immunoblot showing EPRS and Gag expression levels in producer cells. (**B**) p24 level in supernatants, measured by HIV-1 p24 ELISA. (**C**) Viral titer measured by limited dilution in GHOST indicator cells. (**D**) Normalized progeny virion infectivity. Five trials were performed, with the mean value indicated by a horizontal line.

## 4. Discussion

The MA domain of HIV-1 Gag is an RNA-binding protein with a preference for binding to a variety of tRNAs in cells [30]; tRNA–MA interactions have been shown to regulate Gag-membrane association in a wide variety of in vitro and cell-based studies [21,54]. A recent crystal structure of the MA-tRNA^Lys3^ complex showed that the highly basic region (HBR) of MA interacts with tRNA at the elbow region [34]. In addition to tRNA^Lys3^, MA also interacted with tRNA^Gly^, tRNA^Phe^, tRNA^Thr^, tRNA^Leu^, and tRNA^Ser^ in vitro, albeit with lower affinity than tRNA^Lys3^ [34]. In vitro binding and in silico docking studies also suggested that MA can form a stable complex with tRNA^Pro^, and the MA–tRNA interface involved the MA HBR and tRNA D-arm, variable loop, and anticodon arm [33]. The diversity of MA–tRNA interactions raises the possibility that Gag–membrane association is regulated by an ensemble of tRNAs in cells. Here, we confirmed that Gag and MA interact with the human MSC, a potential source of tRNAs. We identified the MSC-associated EPRS protein linker region (WHEP domains) as a primary Gag/MA interaction site and showed that this interaction is bridged by RNA. Although the identity of the RNA was not investigated, we hypothesize that it is tRNA.

While ARSs bind specific tRNAs due to the presence of identity elements found primarily in the tRNA acceptor stem and anticodon loop [55,56], WHEP domains in the linker region of EPRS have been shown to interact more generally with RNAs as well as DNA [57,58]. We observed indiscriminate interaction between the EPRS linker and multiple species of tRNAs, which is consistent with electrostatic interactions between the phosphate backbone of tRNAs and positively charged residues in the linker.

We generated and analyzed a number of MA mutants for EPRS binding. The majority of MA variants defective in EPRS interaction contained mutations within the HBR region. Therefore, MA interacts with EPRS via the same surface with which it interacts with PI(4,5)P_2_ at the PM. Although the role of tRNAs in regulating Gag-membrane association is well established [54], tRNAs in cells are almost always associated with components of the translational or tRNA-trafficking machinery and are rarely free [59]. Thus, the tRNAs that mediate viral assembly are likely bound to other proteins prior to their recruitment. In addition to the Gag–MSC interaction, monomeric Gag has also been shown to interact with ribosomes and specific ribosomal proteins [60,61]. Both the ribosome and the MSC are rich sources of potential regulatory tRNAs that Gag may recruit to suppress non-specific membrane interactions. The MSC may provide the tRNAs that regulate or delay the interaction of MA with the membrane, though none of our manipulations of EPRS expression had major effects on virus assembly. EPRS knockdown had little specific effect on the HIV-1 lifecycle; translation of viral genes was impaired, but this was likely a result of reduced global protein translation in response to a deficiency in charged tRNA^Glu^ and tRNA^Pro^. The observation that EPRS knockdown also had little effect on HIV-1 infectivity of progeny virions suggests that HIV-1 Gag does not require the full basal levels of MSC to obtain regulatory tRNAs. Irrespective of the major source of MA-bound tRNA, Gag needs to dissociate from the ribonucleoprotein complex and release the tRNA prior to ultimate membrane association and virion assembly [30]. Although MA binds to some tRNAs with sub-micromolar affinity in vitro, myristoyl group exposure during the so-called myristoyl switch weakens the affinity [31]. Myristoylated MA binds to model membranes containing PI(4,5)P_2_ and cholesterol with similarly high affinity [62]. MA trimerization has also been shown to strongly promote MA binding to PI(4,5)P_2_-containing lipid nanodiscs [63]. Thus, a variety of mechanisms, including lipid composition and Gag-gRNA and Gag–Gag interactions, likely regulate tRNA release and preferential membrane binding [64].

While the linker region of EPRS was sufficient for MA binding, full-length Gag was able to interact with EPRS variants with their linker regions deleted. Thus, the presence of other domains of Gag likely facilitated its interaction with EPRS, bypassing the requirement of the WHEP domains. In addition, progeny virion production and infectivity were not impacted in a cell line stably expressing the EPRS linker region (Appendix A). Thus, while Gag forms a stable interaction with the MSC that involves the EPRS linker region, overexpression of the free linker domain outside the context of the MSC failed to perturb Gag assembly.

One possibility suggested by the data is that the Gag–MSC association, which is mediated by an interaction between the MA HBR and tRNA, provides Gag with a source of regulatory tRNAs that are otherwise sequestered by the protein synthesis and tRNA trafficking machinery. Inhibition of Gag–EPRS interactions through MA domain mutations inhibited VLP formation in some cases, though not all, and significantly reduced the infectivity of progeny virions in other cases. While we cannot rule out the possibility that these effects may be due to disruption of other known functions of the MA HBR, preventing Gag association with the MSC by specific inhibitors may yet prove to have anti-viral activities by affecting Gag functions.

We note that the interaction of Gag with EPRS may have functions outside the regulation of the Gag interaction with membrane and virion assembly. In particular, while EPRS is best known as a subunit of the MSC in charging tRNAs with their cognate amino acids, as mentioned earlier, EPRS is also a component of the GAIT complex [65]. IFN-γ induces the formation of this heterotetrameric complex, which binds to the 3′-UTR of multiple inflammation-related mRNAs and inhibits their translation. The GAIT complex binds to targeted mRNA via the linker domain of EPRS [65,66], the very domain we find interacting with RNAs, and the MA domain of Gag. The interaction of MA with the EPRS linker domain suggests MA may be involved in the regulation of the inflammation response by targeting the GAIT complex. The specifics of the MA–EPRS interaction revealed here will facilitate future probes into the potential impact of the HIV-1 Gag protein in modulating the IFN response to infection.

## Figures and Tables

**Figure 4 viruses-15-00474-f004:**
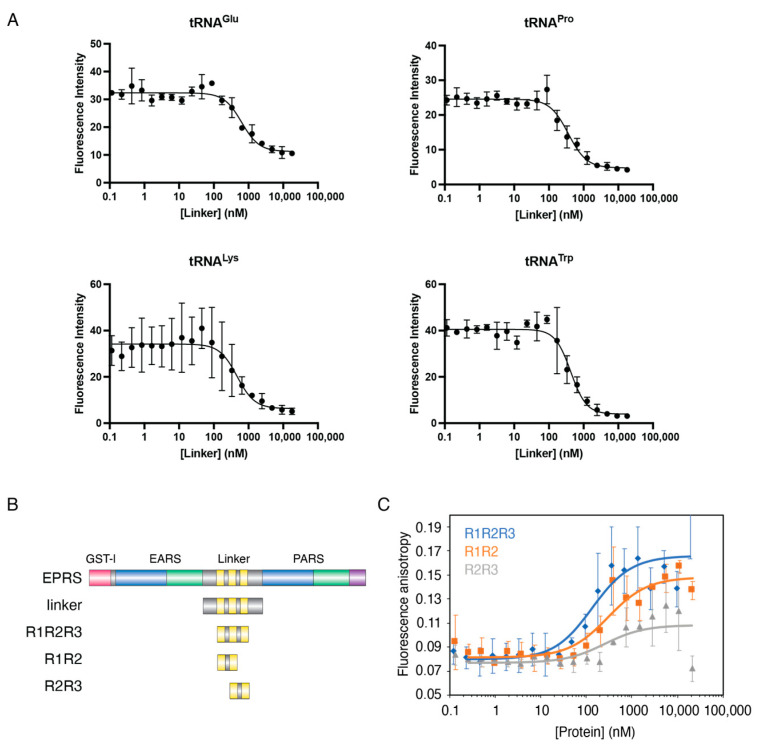
Interaction of EPRS linker with tRNAs in vitro. (**A**) Purified EPRS linker protein was titrated into 5 nM in vitro transcribed, fluorescently labeled tRNA (Glu, Pro, Lys3, or Trp, as indicated). Average fluorescence quenching curves are shown, with the error bars representing the standard deviation of three independent experiments. (**B**) Schematic of full-length EPRS and linker fragments used in this study. (**C**) Graph showing results of FA binding assays wherein purified EPRS linker fragments were titrated into 5 nM tRNA^Lys3^. Data were analyzed as previously described [48]. Each curve is the average of three independent experiments with the standard deviation indicated.

**Table 1 viruses-15-00474-t001:** Apparent binding dissociation constants between tRNA and EPRS linker. Binding assays were performed with 5 nM fluorescently labeled tRNAs and serially diluted proteins in 20 mM Tris-HCl pH 8, 50 mM NaCl, and 1 mM MgCl_2_. The results are the average of three independent trials with standard deviation listed.

Protein	RNA	*K*_d_ (nM)	Method
Linker	Human tRNA^Lys3^	589 ± 152	Fluorescence quenching
	Bovine tRNA^Trp^	511 ± 119	
	Human tRNA^Glu^	886 ± 203	
	Human tRNA^Pro^	449 ± 196	
R1R2R3		43 ± 35	Fluorescence anisotropy
R1R2	Human tRNA^Lys3^	209 ± 58	
R2R3		>10 µM	

## Data Availability

All data collected for this study will be made available upon request.

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
