# Peer review of "HIV-1 Gag Binds the Multi-Aminoacyl-tRNA Synthetase Complex via the EPRS Subunit"

_viruses, 2023, doi:10.3390/v15020474_

Round 1

Reviewer 1 Report

In recent years, it has been found that tRNAs have multiple functions in addition to their canonical role in cellular protein biosynthesis and in priming reverse transcription. The Musier-Forsyth lab has contributed to this new paradigm and for example, previously reported the involvement of the MRC in HIV assembly. In this new paper, a collaboration with the Goff lab, the authors have advanced this area of research in a significant way.  Importantly, they have mapped the primary site of interaction of MA (also the MA domain in Gag) with the MRC to the linker region of Glu-Pro-tRNA synthetase (EPRS) and showed that this interaction is RNA-dependent. In addition, they used alanine-scanning mutagenesis to identify the MA residues that are involved in this interaction. Interestingly, these residues are essential for virus infectivity and also interact with phosphatidylinositol-(4,5)-bisphosphate [PI(4,5)P2], a lipid that ensures the specificity of Gag-membrane interactions.  There are also EPRS knockdown experiments showing that a decrease in virus production is not the result of specific effects on viral gene expression. The paper is well-written and the experiments are well-controlled and convincing. I found the speculation in the last paragraph rather intriguing.  Clearly, a tremendous amount of effort was involved in this project. The paper will be of interest to a wide audience, including investigators working on HIV and other viruses, the tRNA community, and cell biologists interested in membrane interactions with other cellular components.

Although the Introduction is clearly written, the complicated system and the many technical terms that are abbreviated may be challenging for readers not familiar with this field of investigation. I recommend that the authors include a schematic diagram that describes the system to accompany the Introduction and for easy reference, to also include a table within the body of the text that gives a list of all the abbreviations and what they represent.  

Minor comments

1.Abstract, line 18. Define ARS.

2.In Fig. 5, the authors normalized virus production by volume. It would be helpful if they could explain why they felt that this approach is valid. Was this based on the results seen on the blot in Fig. 5A?

3.Minor Typos.

p.6, line 296. “Co-IP’d” is slang. Co-immunoprecipitated should be spelled out.

p.8, line 347. The figure is 3C, not 3B.

p. 14. Fig. 8A. The numbers on the extreme left presumably represent the positions of the marker proteins. The band at 50 kDa should be labeled HIV-1 Gag and the band near 25 kDa should be labeled HIV-1 p24.

p.15, line 586. Insert “to” after “due”.

Reviewer 2 Report

The Jin et al. Viruses manuscript, "HIV-1 Gag binds the multi-aminoacyl-tRNA synthetase complex via the EPRS subunit" appears to be a comprehensive, yet somewhat disappointing, analysis of HIV-1 Gag and matrix (MA) binding to the multi-aminoacyl-tRNA synthetase complex (MAC) via the glutamyl-prolyl-tRNA synthetase (EPRS). In particular, the authors have demonstrated an interaction between the EPRS linker region and the MA phosphatidylinositol-4,5-bisphosphate (PI[4,5]P2) binding region that is mediated via an RNA, presumably a tRNA, bridge. The disappointment is that disruption of this interaction in cell culture doesn't seem to do a whole lot. The analyses of MA mutations that disrupt the EPRS interaction were confounded by the fact that the mutations also affect other MA functions, such as PI(4,5)P2 binding. Similarly, EPRS knockdown reduced HIV-1 gene expression, but via a global translation defect. The authors also observed a very modest effect of EPRS knockdown on HIV-1 virus production, but no effect on virus infectivity. Certainly the observations are worth reporting somewhere, but I wouldn't guess they'll raise a lot of eyebrows. There are also some specific issues that would seem worth attention:

1. Figure 2: It wouldn't hurt to quantify the IP versus input ratios for the EPRS knockdown. I suppose this could go in the figure legend.

2. Figure 3B: I'm uncertain as to why there should be a protein band that is labeled "Strep resin." Please explain.

3. Figure 3C: I was surprised that the experiment was done with Gag instead of MA. More importantly, the complete linker deletion control (instead of just the R1R2R3 deletion) ought to have been included. The hope would be that the full linker deletion would not interact with Gag.

4. Figure 4A and Table 1: I would have thought that using a random RNA control in addition to the tRNAs would have been in order.

5. Figure 4C: Where is the full linker deletion control?

6. Figure 5A: I think readers deserve to be given all the IP versus input ratios for all the MA mutants, which presumably would explain how specific mutations were picked for further analyses.

7. Figure 8A: The authors should let readers know if more cell lysate was used in the shEPRS lanes versus the shN.S. lanes. If EPRS knockdown globally reduces translation, one would expect to see less Gag and less GAPDH in the EPRS knockdown lanes, unless lanes were normalized for GAPDH content. In this regard, if cellular Gag levels were reduced, this might have had a disproportionate effect on virus assembly and release, which is, in part, Gag concentration-dependent.
